# Role of Nitric Oxide in Megakaryocyte Function

**DOI:** 10.3390/ijms24098145

**Published:** 2023-05-02

**Authors:** Amir Asgari, Paul Jurasz

**Affiliations:** 1Faculty of Pharmacy and Pharmaceutical Sciences, University of Alberta, Edmonton, AB T6G-2E1, Canada; 2Department of Pharmacology, University of Alberta, Edmonton, AB T6G-2H7, Canada; 3Cardiovascular Research Institute, University of Alberta, Edmonton, AB T6G-2S2, Canada; 4Mazankowski Alberta Heart Institute, Edmonton, AB T6G-2R7, Canada

**Keywords:** megakaryocyte, platelet, nitric oxide, endothelial nitric oxide synthase, inducible nitric oxide synthase

## Abstract

Megakaryocytes are the main members of the hematopoietic system responsible for regulating vascular homeostasis through their progeny platelets, which are generally known for maintaining hemostasis. Megakaryocytes are characterized as large polyploid cells that reside in the bone marrow but may also circulate in the vasculature. They are generated directly or through a multi-lineage commitment step from the most primitive progenitor or Hematopoietic Stem Cells (HSCs) in a process called “megakaryopoiesis”. Immature megakaryocytes enter a complicated development process defined as “thrombopoiesis” that ultimately results in the release of extended protrusions called proplatelets into bone marrow sinusoidal or lung microvessels. One of the main mediators that play an important modulatory role in hematopoiesis and hemostasis is nitric oxide (NO), a free radical gas produced by three isoforms of nitric oxide synthase within the mammalian cells. In this review, we summarize the effect of NO and its signaling on megakaryopoiesis and thrombopoiesis under both physiological and pathophysiological conditions.

## 1. Basic Megakaryocyte Biology

Megakaryocytes are large (50–100 µM), multi-nucleated cells that are responsible for releasing platelets into the blood [1]. They are characterized by a multilobulated nucleus accumulating 2n, 4n, 8n, 16n, 32n, up to 128n DNA content and constitute ~0.01% of bone marrow cells [2]. Recent evidence sheds light on the different roles of megakaryocytes in various physiological and pathophysiological processes. Until recently, it had been thought that megakaryocytes solely serve as progenitors or precursor cells responsible for platelet production. This simple notion may no longer be valid as recent reports have uncovered various roles of megakaryocytes in the immune response and in modulating the proliferation and differentiation of different cell lineages, particularly osteoblasts and osteoclasts within the bone marrow [3,4,5]. Megakaryocytes are capable of antigen endocytosis and, ultimately, its presentation within MHC I to CD8^+^ T cells [6]. Moreover, recent studies revealed that megakaryocytes release several immune-modulatory cytokines, including TGF-β and IL-1, and express co-stimulatory molecules such as CD40L and B7-2 (CD86) on their surface, suggesting they act as antigen presenting cells (APCs) within the bone marrow microenvironment [5,7,8,9,10]. Of interesting note, recent evidence suggests that megakaryocytes may act as the first line of defense against cancer metastasis to the bone [3,4]. Therefore, like platelets, which are increasingly recognized for their diverse roles beyond hemostasis [11,12,13,14], megakaryocytes may have functions beyond platelet production. As such, a greater understanding of how important chemical mediators influence megakaryocytes in platelet production and newly recognized functions is required.

According to conventional or classic hematopoiesis, hematopoietic stem cells (HSCs) give rise to megakaryocyte-biased progenitors after passing through several strict commitment points or lineage-biased steps like a hierarchical-branched tree [15,16]. However, more recent evidence demonstrates that although hematopoietic stem cells are capable of reconstituting all blood cell lineages, they may exhibit megakaryocyte or platelet-biased phenotypic and functional characteristics. Therefore, these multipotent progenitor cells may bypass differentiation pathways and directly give rise to megakaryocyte or platelet-committed progenitors at a very early step in differentiation [17,18,19,20,21,22,23]. Consistently, bone marrow transplantation in humans demonstrates that platelet reconstitution takes place earlier than that of other blood cell lineages [24].

It has been reported that HSC differentiation toward megakaryocytes and their maturation, endomitosis, and invaginated membrane system (IMS) development takes place in the osteoblastic niche, whereas a later generation of proplatelets, megakaryocytye-platelet intermediate pseudopodia-like structures, requires vascular niche localization [2]. Within these bone marrow niches HSC/megakaryocyte progenitor cell interactions with microenvironment extracellular matrix proteins help to regulate megakaryocyte differentiation and platelet production. HSC interaction with type I collagen within the osteoblastic niche via VLA-2 (integrin α_2_β_1_) promotes commitment to megakaryocyte-biased progenitor formation and maturation, but suppresses megakaryocyte terminal development, which results in proplatelet generation [25,26,27]. Similarly, megakaryocyte glycoprotein (GP) VI–collagen I interaction has inhibited proplatelet formation [28]. However, double knockout of collagen receptors (GPVI−/− integrin α_2_β_1_−/−) shows no difference in megakaryocyte distribution, size, or blood platelet levels compared to that of wild type mice, suggesting other regulatory mechanisms may exist to suppress ectopic proplatelet generation within the osteoblastic niche [29]. In contrast, the vascular niche contains extracellular matrix proteins including collagen IV, fibronectin, fibrinogen, and von Willebrand factor, which induce proplatelet generation [30,31,32,33]. Other factors, including megakaryocyte-active mitogens such as fibroblast growth factor 4 (FGF-4) and the chemokine stromal cell-derived factor 1 (SDF-1) also promote survival, maturation, and platelet production from megakaryocytes by facilitating their chemotaxis toward and affinity for bone marrow sinusoid endothelial cells [34,35]. Once in the vascular niche, several hypotheses have been proposed to explain the mechanism behind the proplatelet extension from megakaryocytes into the lumen of bone marrow blood vessels [36,37,38]. A concentration gradient of sphingosine-1 phosphate (S1P) has been shown to exist at the contact site between the megakaryocytes and sinusoidal blood, which directs proplatelets into lumens in a sphingosine-1-phosphate receptor 1 (S1prP1)-dependent manner [39]. Ultimately, blood flow shear forces facilitate proplatelet release from the megakaryocyte and their fission to produce platelets.

In addition to extracellular matrix proteins, various soluble factors have been proposed to play important roles in regulating megakaryopoiesis and thrombopoiesis. Of these, particularly important is the glycoprotein hormone thrombopoietin (TPO). Through its receptor c-mpl, which is expressed on the most primitive HSCs, TPO plays a key role in megakaryocyte differentiation from HSCs and their maturation toward platelet generation [15,40]. TPO plays a central role in maintaining platelet/megakaryocyte-biased HSCs, as TPO knockout (TPO−/−) bone marrow cells give rise to lymphoid-biased bone marrow reconstitution in irradiated recipient mice [18]. As such c-mpl and TPO knockout mice demonstrate 90% reductions in megakaryocyte and platelet numbers [41], while loss of function mutations to Mpl within humans cause congenital amegakaryocytic thrombocytopenia, resulting in a severe phenotype only rescued by bone marrow transplantation. In addition to TPO, several other factors have been identified which promote megakaryocyte proliferation and maturation, including interleukin 3(IL-3), interleukin 6(IL-6), and stem cell factor (SCF) [42,43,44].

TPO also induces megakaryocyte polyploidization, which results in the accumulation of lipids and proteins required for the constitution of a vast invaginated membrane network connected to the megakaryocyte surface membrane. This membrane network forms the surface membrane of proplatelets and the cytoskeletal ultrastructure that supports the elongation of proplatelet tubular structures [1,45,46,47]. The process of proplatelet formation and the release of platelets into the sinusoidal blood vessels in the bone marrow is highly regulated [48], During this process, cytoskeletal proteins, including β1-tubulin, dynein, F-actin, and myosin II, facilitate proplatelet generation by providing assembly lines for elongation, organelle transportation, and ultimately platelet release [35,47,49,50]. Of particular importance is the role of the transcription factor NF-E2 and its expression of β1-tubulin that plays a pivotal role in proplatelet biogenesis, structure, and function by polymerizing into microtubule bundles and coils that extend throughout these cytoplasmic extrusions. Consequently, NF-E2 and β1-tubulin knockout mice suffer from thrombocytopenia because of a significant reduction in proplatelet formation [49,51,52,53,54,55]. TPO also induces reactive oxygen species (ROS) production, which play an important role in driving HSC differentiation toward mature megakaryocytes and platelet production. This ROS generation likely involves nicotinamide-adenine dinucleotide phosphate (NADPH) oxidases (NOXs) and increased oxygen tension, resulting in enhanced tyrosine phosphorylation, proliferation, and polyploidization [56,57]. Moreover, NF-E2 in addition to expressing platelet genes also maintains a moderate expression of cytoprotective genes allowing for ROS accumulation during megakaryocytic maturation [58]. The initiation of platelet formation from mature megakaryocytes has also been shown to be governed by a reciprocal interplay between mitochondrial dynamics and ROS, in which increased ROS levels stimulate mitochondrial fission, leading to the production of more mitochondrial ROS [59]. Most recently, a role for ROS has also been identified in the pulling of megakaryocyte intravascular proplatelet extensions by so-called “plucking” neutrophils to enhance platelet formation [60].

However, it is worth noting that platelet generation via megakaryocyte proplatelet formation at steady state may differ mechanistically from platelet production in response to stress or injury. Stress thrombopoiesis, or the process of platelet production under inflammatory or acute thrombocytopenia conditions, can occur much faster than physiological platelet production [61,62]. This may occur in part due to the presence of platelet- or megakaryocyte-primed hematopoietic stem cells (HSCs) in the bone marrow that can bypass the traditional route of multi-step lineage-biased progenitor differentiation and give rise to platelets more quickly [18,19,23,62,63]. However, equally important to stress thrombopoiesis is whether platelet generation proceeds through or bypasses the need for TPO and classic proplatelet formation. Although proinflammatory cytokines, such as IL-1β, can upregulate the expression of TPO and other megakaryocyte-related transcription factors to further promote platelet production [64], recent studies have shown that in response to IL-1α megakaryocytes undergo rupture to rapidly produce platelets in a TPO-independent manner after platelet loss or inflammatory stimulus [65,66]. This rupture-dependent thrombopoiesis also displays caspase-3 dependence [65], and platelet generation differences at stress vs. steady state may help to explain whether or not megakaryocyte apoptosis needs to be restrained for platelet generation and which apoptotic pathways may or may not be involved [67,68,69,70].

While much is known about the roles of extracellular matrix proteins, soluble protein mediators, and even gaseous chemical mediators such as ROS in megakaryopoiesis and thrombopoiesis, relatively little is known of the role of nitric oxide (NO) in these processes. This is somewhat surprising considering NO’s pleiotropic biological activity, and the important role it plays regulating hematopoiesis and platelet function [71,72,73,74,75]. Therefore, in this mini review we summarize the role of NO and its signaling on megakaryocyte function.

## 2. Basic Nitric Oxide Biology

Nitric oxide (NO) is a highly diffusible free radical gas with a short half-life that plays an important role in many physiological and pathophysiological processes [76], including regulating vascular tone and signal transmission by neurons [77,78,79,80,81]. Importantly, it also plays a major role in immune function as well as within the hematopoietic system [82,83,84,85,86].

NO is produced enzymatically from the oxidation of L-arginine by NADPH-dependent family oxidation-reduction enzymes called nitric oxide synthases or NOSs [87,88,89]. These enzymes utilize flavin adenine dinucleotide (FAD), flavin mononucleotide (FMN), and (6R-)5,6,7,8-tetrahydro-L-biopterin (BH4) as cofactors to generate NO from the substrate L-arginine and co-substrates oxygen and NADPH. Three isoforms of nitric oxide synthase exist, including NOS I (nNOS, neuronal nitric oxide synthase), NOS II (iNOS, inducible nitric oxide synthase), and NOS III (eNOS, endothelial nitric oxide synthase) (Figure 1) [90,91]. Although all three enzymes bind to calmodulin (CaM), nNOS and eNOS bind CaM upon a rise in intracellular Ca^2+^ concentration and become activated [92,93,94]. Of further importance for eNOS regulation is its localization to cell membrane caveole wherein the caveolae coat protein caveolin-1 is a tonic inhibitor of eNOS activity and recruitment of CaM and heat shock protein 90 displaces caveolin-1, leading to eNOS activation [95,96]. eNOS activity is also widely regulated both positively and negatively via phosphorylation, with Ser1177 and Thr495 being the most widely studied of such sites. eNOS activating phosphorylation occurs in response to circulating mediators such as vascular endothelial growth factor, insulin, bradykinin, and estrogen, as well as in response to sheer stress [97]. Similar to constitutive NOS enzymes, iNOS also binds to calmodulin; however, it does so even at basal levels of intracellular Ca2+ due to its high affinity for CaM [98]. nNOS and eNOS are constitutively expressed in different cells and tissues; while, under physiological conditions iNOS expression is limited [99,100,101], but can be induced in almost any cell type by proinflammatory proteins such as IL-1, TNF-α, IFN-γ, IL-2, IL-12, IL-18, CD40 ligand and Fas-ligand and pathogen-associated molecular patterns such as lipopolysaccharide [101].

NO exerts most of its biological functions through interaction with various key regulatory proteins either via direct binding to targets (cGMP-independent effects) or via cGMP-dependent signaling following its activation of soluble guanylate cyclase (sGC). Activation of sGC and cGMP generation can result in activation of cyclic nucleotide-gated ion channels as well as protein kinase G (PKG) activation and signaling, which can modulate diverse cellular processes such as regulation of enzyme activity, gene transcription, and post-translational modification [80,102,103,104,105,106]. Conversely, NO may bind to heme and regulate the activity of heme-containing enzymes such as cytochrome C oxidase or other protein/peptides via nitrosylation of thiol groups to regulate important processes such as apoptosis [97].

NO may be inactivated in a number of ways including by reacting with the heme in deoxyhemoglobin to form a stable complex and in the presence of oxygen to form methemoglobin and nitrate. It also reacts with oxygen and water to yield nitrite and nitrate in a series of reactions with a dinitrogen tetraoxide intermediate. NO also rapidly reacts with superoxide (O_2_^−^) to form peroxynitrite (ONOO^−^), a highly active and cytotoxic radical, which subsequently reduces NO bioavailability [107,108,109]. In high concentrations ONOO^−^ may exert proapoptotic and cytotoxic effects through various mechanisms, including protein oxidation and tyrosine nitration, lipid peroxidation, disruption of the electron transport chain and TCA, and via single-strand breaks in DNA [109,110,111,112,113,114,115,116].

## 3. Nitric Oxide and Platelet Function

In the 1980s, Radomski and Moncada demonstrated that NO potently inhibits platelet adhesion and aggregation [117,118,119,120]. Subsequently, NOS and an NO signaling pathway were identified within platelets [75,121,122,123], and NO produced during aggregation was shown to inhibit further platelet recruitment [124,125]. NO mediates most of its platelet inhibitory effects via cGMP generated by sGC (Figure 2) [126,127,128]. cGMP acts on PKG, which phosphorylates vasodilator-stimulated phosphoprotein (VASP), enabling VASP binding to the platelet cytoskeleton [129,130]. Next, VASP inhibits integrin α_IIb_β_3_ activation, preventing adhesion and aggregation [131,132]. PKG signaling is also reported to suppress intracellular Ca^2+^ and integrin α_IIb_β_3_ activation via inositol-1,4,5-triphosphate receptor-associated cGMP kinase substrate signaling [133,134] and to suppress thromboxane receptor activation [135]. Platelet NOS activity has been attributed to eNOS [121,136,137,138], although a few studies report iNOS in low amounts in platelets [136,137] (there are no reports of platelet nNOS).

In the past 20 years, however, controversies have arisen over platelet NO signaling. The most relevant questioned platelet NO production and eNOS presence [139] and whether NO also has a stimulatory role in platelet activation [140]. To address these controversies, we previously investigated the hypothesis that some of these discrepancies may be explained by differences in platelet levels with and without eNOS signaling. Recently, we identified eNOS*^neg/low^* and eNOS*^pos/high^* platelet subpopulations in blood [141]. We demonstrated that eNOS*^neg/low^* platelets do not produce NO or produce it in low amounts. This platelet subpopulation also has a down-regulated sGC-PKG-VASP signaling pathway, initiates adhesion to collagen, and more readily activates integrin α_IIb_β_3_ than eNOS*^pos/high^* platelets. eNOS*^pos/high^* platelets contain higher protein levels of sGC, PKG, and VASP and are more abundant (~80% of total platelets). eNOS*^pos/high^* platelets also form the bulk of an aggregate via enhanced COX-1 signaling; however, they also ultimately limit aggregate size via NO generation.

Importantly, ONOO^−^ also impacts platelet function [142], and its impacts may help explain some of the discrepant findings surrounding platelet NO function. At low concentrations, peroxynitrite was shown to mediate NO-dependent platelet inhibition; however, at higher concentrations it caused an increase in P-selectin exposure and platelet activation [143]. Consistently reducing peroxynitrite formation by suppressing NADPH oxidase, a major source of platelet superoxide generation, was shown to increase NO bioavailability and subsequent platelet inhibition [144].

Insufficient platelet NO production and a decrease in its bioavailability may also have important pathological consequences, particularly in the setting of acute coronary syndrome (ACS). Platelets from ACS patients have impaired NO production [145], and platelet NO production inversely correlates with increasing number of coronary artery disease risk factors [146]. Similarly, platelet refractoriness to the NO donor sodium nitroprusside predicts increased morbidity and mortality in patients with high-risk ACS [147]. Consistent with these findings, megakaryocytes from patients with normal coronary arteries have been reported to generate more NO in a Ca^2+^-dependent manner than megakaryocytes from patients with atherosclerosis, although megakaryocytes from atherosclerotic patients generate more NO in an iNOS-dependent manner [148,149,150]. That platelets have a more limited transcriptome and capacity for new protein synthesis and that iNOS protein has an extremely short half-life (<2 h) [151,152] suggests that reduced platelet NO bioavailability within coronary artery disease may reflect a reduction in megakaryocyte eNOS expression. Furthermore, NO formed from different NOS isoforms may play differing roles in megakaryocyte vs. platelet function. Hence, due to recent advances in our understanding of platelet NO biology and its significance to pathology, a closer examination of NO-signaling in megakaryocytes is also needed.

## 4. Nitric Oxide Synthases in Megakaryocytes

As described above, constitutive (Ca^2+^-dependent) and inducible NOS isoforms have been identified in both human bone marrow megakaryocytes [148] and within the Meg-01 megakaryoblastic cell line [153]. Treatment of Meg-01 with proinflammatory cytokines IL-1β and TNF-α also revealed a reciprocal relation between constitutive and inducible NOS activity consistent with an increase in iNOS expression and a down-regulation of constitutive NOS expression. The Ca^2+^-dependent NOS in megakaryocytes/blasts likely corresponds to eNOS as its expression has been confirmed via RT-PCR and immunostaining within Meg-01 [141]. Moreover, like in platelets, both eNOS*^neg/low^* and eNOS*^pos/high^* Meg-01 subpopulations have been identified [141,154].

## 5. Effect of NO on Differentiation and Proliferation of Megakaryocytes

Early research demonstrated that high concentrations (μM) of the NO donor DETA/NO induce apoptosis of bone marrow-derived CD34^+^ progenitor cells and that iNOS-generated NO may in part mediate hematopoietic suppression by proinflammatory cytokines IFN-γ and TNF-α [71]. Treatment of human bone marrow-derived and TPO-cultured CD34^+^ cells, as well as mononuclear cells, with IFN-γ and TNF-α reduces the number of CD41^+^ cells after 12 days of culture, while high NO-donor concentrations inhibit the outgrowth of megakaryocytes derived from these cells by inducing their apoptosis [86]. Prostacyclin treatment and cAMP signaling protect megakaryocytes outgrown from CD34^+^ cells from NO-induced apoptosis [155], while TPO, 5-hydroxytryptamine, and IL-11 appear to protect megakaryocytic cell lines from apoptosis induced by high NO concentrations achieved by NO donors or iNOS induction [84]. Altogether, these results suggest that in absence of protective factors and under inflammatory-like conditions up-regulation of iNOS expression and increased NO concentrations induce apoptosis of progenitor cells, preventing their differentiation toward megakaryocytes. Whether the pro-apoptotic effects of NO are mediated via cGMP-dependent or non-cGMP-dependent mechanisms remains to be fully elucidated, as does the contribution of peroxynitrite and of the apoptotic pathways involved. Moreover, potential cross-talk between pathways that retard NO-induced apoptosis of megakaryocytes and their progenitors also needs further investigation [156].

## 6. Effect of NO on Platelet Production by Megakaryocytes

Similar to the limited number of studies investigating the role of NO in megakaryopoiesis, there is also a paucity of data with regards to NO’s role in thrombopoiesis. Early work by Loscalzo and colleagues demonstrated that treatment of the Meg-01 cell line with high NO concentrations as achieved by utilizing the NO-donor S-nitrosoglutathione (GSNO) or by treating the Meg-01 cell line with proinflammatory cytokines (IFN-γ, TNF-α, and IL-1β) induces the generation of CD41^+^ platelet-sized particles in culture with a capacity to aggregate [85]. Moreover, platelet particle generation by Meg-01 is further enhanced if the Meg-01 are pretreated with TPO prior to stimulation, although TPO treatment alone was not able to promote platelet particle generation consistent with its role in megakaryocyte maturation [85]. The mechanism by which high NO concentrations induce platelet-sized particle formation was reported to be cGMP-independent, and interestingly, was associated with the generation of distinct Meg-01-derived annexin-V and propidium iodide positive apoptotic bodies. This finding led the authors to hypothesize that NO-induced apoptosis is related to the process by which megakaryocytes produce platelets, although as also noted by the authors it is not clear whether the observed apoptosis is a result of removal of spent megakaryocytes or apoptosis and platelet production are simultaneous events [84,85]. Lastly, of note, the authors identified that iNOS null mice demonstrate platelet counts nearly half of that of their wild-type or eNOS null counterparts, further exemplifying the important role of NO in platelet production.

Consistent with the findings of Loscalzo and colleagues, intravenous infusion of L-nitroarginine or N(G)-nitro-L-arginine methyl ester, both NOS inhibitors, to rats results in thrombocytopenia or decreased platelet counts [157]. More recently, CD226 whole body or platelet/megakaryocyte specific knockout mice have been shown to have elevated platelet and megakaryocyte (bone marrow and spleen) counts compared to wild-type controls [158]. Notably, the platelets from CD226−/− mice demonstrated greater aggregation response to thrombin compared to platelets from WT mice, attributed to their reduced eNOS levels and decreased ability to generate NO. Currently, it is unknown whether the potential alterations in megakaryocyte-platelet NO-signaling in these mice impact their megakaryo- or thrombopoiesis. However, considering that platelet function may be regulated by low NO concentrations attributable to eNOS activity while high NO concentrations associated with iNOS appear to have profound effects on megakaryocytes and their potential to produce platelets, it is tempting to speculate whether these two NOS isoforms have differential function in megakaryocytes vs. platelets. Specifically, the role of iNOS and its increased expression may be of particular importance to platelet production under stress such as in the case of rupture thrombopoiesis as it can be rapidly by induced by IL-1α [159] or in cases of inflammation/infection-induced secondary (reactive) thrombocytosis (Figure 3).

## 7. Summary and Conclusions

Although the role of NO in platelet biology is well studied, much less is known about its effects on megakaryocytes. To date, most studies have focused on the ability of NO at high concentrations to promote cell death of megakaryoctyes and their progenitors. Considering NO’s pleiotropic effects and its ability to influence HSC mobilization [160], future studies may need to focus on its role in regulating megakaryocyte progenitor interaction with their microenvironments within the osteoblastic and vascular niches as well as its impact on ROS-mediated signaling during megakaryocyte differentiation. Further, previous studies focusing on NO’s impact on platelet production have modeled conditions of stress rather than steady-state thrombopoiesis, implicating a role for iNOS in this process. Indeed, analogously increased iNOS expression and NO generation have recently been identified to play a role in stress erythropoiesis [161]. Considering that NO may have both pro- and anti-apoptotic roles [162], future studies delineating the role of NOS isoforms and NO in steady-state vs. stress thrombopoiesis may also clarify whether apoptosis plays a role and what role it plays in platelet production [163,164]. Lastly, further studies are also warranted to identify whether NOS-based subpopulations of megakaryocytes exist as they do for platelets [141]. The results of such studies may shed new light on the development of novel genetic and/or pharmacological tools in order to manipulate NO-signaling within megakaryocytes and their platelet progeny for therapeutic purposes.

## Figures and Tables

**Figure 1 ijms-24-08145-f001:**
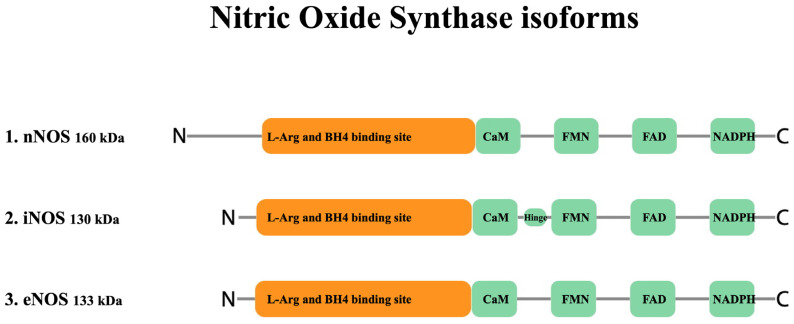
Structure of nitric oxide synthase enzyme isoforms. Both nNOS (NOS I) and eNOS (NOS III) are capable of synthesizing NO in a short pulsative manner ranging from pM to nM (low) and nM to μM (moderate) concentrations, respectively. However, iNOS (NOS II) synthesizes a significantly higher amount of NO in the range of μM concentration, in a constant manner after the expression of the enzyme. (BH4: (6R-)5,6,7,8-tetrahydro-L-biopterin, a co-factor essential for NOS activity.)

**Figure 2 ijms-24-08145-f002:**
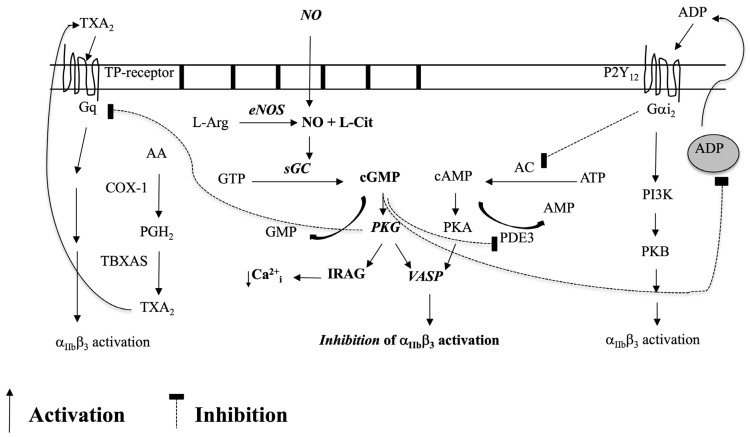
Platelet eNOS-NO-sGC-PKG-signaling and its cross-talk with the PGI2/cAMP, COX-1/TXA2, and ADP/P2Y12 pathways.

**Figure 3 ijms-24-08145-f003:**
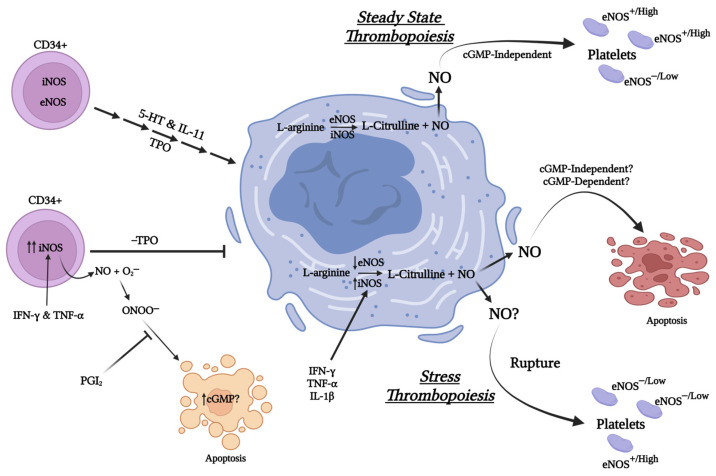
Cartoon summarizing the impact of Nitric oxide derived from iNOS and eNOS on megakaryopoiesis and thrombopoiesis under stressed and non-stressed conditions. (Created with BioRender.com, accessed on 28 April 2023).

## Data Availability

Not applicable.

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
