# Peer review of "Role of Nitric Oxide in Megakaryocyte Function"

_ijms, 2023, doi:10.3390/ijms24098145_

Round 1

Reviewer 1 Report

In this manuscript by Amir Asgari and Paul Jurasz, an overview of the role of nitric oxide in the biology of megakaryocytes (MK) is provided. The work would be an interesting read for IJMS readership, but the manuscript in its current form is still in the draft stage and requires restructuring and extensive editing before resubmission. The authors should distinguish between important and less important messages, choose the references accordingly and clearly focus their work.

The major issue with this work is the poor structure and the verbose writing style of the authors. Usually, a review manuscript starts with a clear outline on MK biology (in this case), with the prerequisites for biogenesis and function, then gradually "funneling" to the title topic (here NO in MK function). Instead, the introduction (and the rest of the work) is rather detailed, listing study after study and thus remaining rather unclear. After the introduction, a sudden "Pharming" paragraph is introduced, which has little to do with the general topic (yet interesting). This could be included much later and should also include the use of microfluidics (flow appears to be essential for proper platelet formation).

The growth factor part in thrombopoiesis can be much clearer. In the next paragraph, we learn that TPO is important for ploidy (line 128), while in line 118 we get the impression that TPO is not so important. A structured intro (see above) on MK maturation and the basic signals would help.

The remark on blood flow in line 157 comes too late and should be part of an introduction.

Figure 2: What is BH4? This should be explained along with NOS.

Between lines 198 and 240 it is unclear whether is spoken of platelets, megakaryocytes, physiology or pathophysiology. The focus appears to move throughout these paragraphs.

It is commendable that the authors address thrombopoiesis during inflammation. There is an interesting new paper on this topic worth addressing (Pubmed ID: 36272416).

Lines 241 to 403 (end of body text) can be shortened quite considerable. There tend to be repetitions throughout the text (lines 231, 333, 337, 345) about the role of NO and ONOO- in platelets, with a duplicate reference accordingly (77,147). The reader is left unclear what ONOO- is for MK (good or bad).

Throughout the manuscript: wording needs to be more accurate. Factor VIII is not a prothrombotic factor, but rather a procoagulant factor. Singular is often used where plural is more appropriate. These small errors add up and tend to distract the reader from the paper's message.

The references are inconsistent, quite old (many before 2000) and there are too many. The authors should distinguish details from main affairs. 

Reviewer 2 Report

The authors should address these comments.

1. In order to better understand, the authors should consider the nitric oxide signaling pathway in megakaryocytes and platelets as a figure.

2. The author should point to the effects of insufficient nitric oxide on platelet activation and function.

3. It is necessary to consider the role of nitric oxide in platelet disorders and  vascular disease in a separate section.

4. Besides nitric oxide, other ROSs play a role in the function of megakaryocytes, which should also be mentioned.

5. How does the interaction of megakaryocytes with lymphocytes in the presence of nitric oxide lead to the maturation and proliferation of megakaryocytes?

6. The authors should have mentioned the pharmacological role of nitric oxide in platelet function in a separate section.

Round 2

Reviewer 1 Report

I thank the authors for taking up my comments so thoroughly. The paper has improved considerably and I endorse publication now.